# Two-colour light activated covalent bond formation

Sarah L. Walden[1,2,4], Leona L. Rodrigues[1,2,4], Jessica Alves [1,2], James P. Blinco [1,2✉], Vinh X. Truong [1,2✉] & Christopher Barner-Kowollik [1,2,3✉]

We introduce a photochemical bond forming system, where two colours of light are required to trigger covalent bond formation. Specifically, we exploit a visible light *cis/trans* isomerization of chlorinated azobenzene, which can only undergo reaction with a photochemically generated ketene in its *cis* state. Detailed photophysical mapping of the reaction efficiencies at a wide range of monochromatic wavelengths revealed the optimum irradiation conditions. Subsequent small molecule and polymer ligation experiments illustrated that only the application of both colours of light affords the reaction product. We further extend the functionality to a photo reversible ketene moiety and translate the concept into material science. The presented reaction system holds promise to be employed as a two-colour resist.

[1] Centre for Materials Science, Queensland University of Technology (QUT), 2 George Street, Brisbane, QLD 4000, Australia. [2] School of Physics and Chemistry, Queensland University of Technology (QUT), 2 George Street, Brisbane, QLD 4000, Australia. [3] Institute of Nanotechnology (INT), Karlsruhe Institute of Technology (KIT), Hermann-von-Helmholtz-Platz 1, 76344 Eggenstein-Leopoldshafen, Germany. [4]These authors contributed equally: Sarah L. Walden, Leona L. Rodrigues. ✉email: j.blinco@qut.edu.au; vx.truong@qut.edu.au; christopher.barnerkowollik@qut.edu.au

Photochemistry has undergone extensive advancement within the last few decades, now providing a powerful synthetic tool for the design of (macro)molecules with advanced architectures and properties. The unique spatial and temporal control provided by light enables a range of applications extending from surface patterning[1–5], to polymer network formation[6–10], and modulation of biological properties[11–14]. Advances in photochemistry, coupled with modern laser instrumentation, further enables the use of distinct monochromatic wavelengths to conduct highly orthogonal reactions in complex chemical environments, known as $\lambda$-orthogonal reactivity[15–18]. Through this innovation, well-defined control over the photochemical transformations can be attained by regulating the activation and deactivation of reactions with two independent wavelengths of light which serve as gates for a single reaction output. The emergence of such dual-wavelength photochemical systems opens up new avenues for novel lithographic techniques[19–21], and the fabrication of materials with customised micro- or nano-structures[22,23].

Dual-wavelength photochemistry requires the design of two photoreactive compounds which (i) possess a compatible $\lambda$-orthogonal window and (ii) are of reversible nature to avoid competing side reactions. To date, there are very few reports on such photochemical systems. The team of Hecht recently introduced a dual-colour photoinitiator for a new method of volumetric two-colour 3D printing technology, referred to as xolography[21]. The employed system integrates a benzophenone type II photoinitiator into a spiropyran photoswitch, enabling initial excitation (spiropyran to merocyanine) and subsequent benzophenone radical generation at 585 nm and 375 nm, respectively. While this method satisfies criteria (ii), the merocyanine also absorbs light in the UV region, leading to competing initiation[21]. Thus, to support the development of advanced two-colour printing systems, there exists a critical need to develop photoresist systems that are strictly gated by two colours of light, ideally enabled by reversible reactive states of two independent chromophores. Our team recently combined visible light-enabled deprotection of dithioacetal-protected aldehyde moieties with UV light-triggered photo-enol ligation for a dual-colour reaction system[24]. However, the photo-deprotection step requires a metal catalyst and is not reversible, which limits its practical application. An alternative system by Villabona et al.[25] employed a UV-activated diarylethene photoswitch, which can be deactivated with red light, inhibiting its reaction with the photo-enol. This system however, does not strictly require two wavelengths, as it can be activated by only blue/UV light.

A major challenge in dual-wavelength photochemistry is to completely inhibit the reactivity of the red-shifted chromophore in the shorter wavelength region, where the other photoreactive moiety is activated. This is non-trivial, as most visible light-absorbing chromophores also have absorbance bands extending into the UV region. Until now, the design of a photoresist that requires two disparate colours of light to form well-defined and distinct covalent bonds has not been reported. Development of such a system, as introduced herein, will significantly broaden the range of chemical and mechanical properties that can be achieved in advanced lithographic techniques, such as dual-colour 3D printing.

Herein, we report a dual-colour photoligation system based on a unique combination of the azobenzene photoswitching and the reversible photogeneration of a reactive ketene[26,27]. Our pioneered photochemical system utilises a redshifted *ortho*-substituted tetrachloride azobenzene moiety and a thermally stable cyclobutenedione[28] that can be activated by visible and UV light, respectively (Fig. 1). Critically, a ligated product can form only when both chromophores are activated by their respective wavelengths. Using action plot studies, we thoroughly investigate the photochemical properties of each photoreactive moiety, revealing the optimum wavelengths for two-colour activation. We have fully characterised the ligated products in both small molecule reaction and polymer endgroup modification. Furthermore, we have developed a resin formulation that can be cured exclusively by two wavelengths of light, demonstrating the potential of our photochemical system in dual-colour 3D printing.

## Results

**Assessment of the $\lambda$-orthogonal activation windows.** Since most photoswitches absorb light across a range of wavelengths, and photoreactivity is often not congruent with the absorption spectrum[29], it is essential to determine the activation regions where no overlapping of the reactivity occurs in dual-colour photochemical systems. Here, we investigated the photochemical properties of two molecules: a redshifted *ortho*-substituted tetrachloride azobenzene **A1** and an $\alpha$-diazo ketone **K1**, which can be photo-converted to a reactive ketene **K2** (Fig. 2). The optimal wavelength for the activation of each photochemical moiety was elucidated via wavelength resolved screening with a nanosecond pulsed, tuneable optical parametric oscillator (OPO) laser. The results are presented in the form of action plots[29], displaying the wavelength-dependent reactivity relative to the absorption spectrum (Fig. 2). The wavelength-dependent *trans/cis* isomerisation of **A1** was determined using a laser-coupled in situ spectrometer (refer to Supplementary Information Section 1.9). Samples of *trans*-**A1** were prepared in dichloromethane (DCM) and irradiated with various wavelengths between $350 < \lambda < 650$ nm. The switching rate from *trans*-**A1** to *cis*-**A1**, presented in Fig. 2A (filled squares), was determined using an exponential fit and normalised by the laser power at each wavelength. This procedure was repeated for the switching of *cis*-**A1** to *trans*-**A1** and the result is presented in Fig. 2A (open circles). Tracking of in situ absorbance measurements during irradiation also afforded the equilibrium between the two isomers for each wavelength (refer to Supplementary Information Fig. 45). For wavelengths $\lambda < 550$ nm, an equilibrium between the two states was produced when starting from *trans*-**A1**. For complete isomerisation, wavelengths $\lambda \geq 550$ nm were required. At $\lambda \approx 375$ nm an equilibrium was obtained where negligible switching between the two isomers was observed, affording a narrow window for the $\lambda$-orthogonal activation of the ketene. As expected, this wavelength corresponds to the isosbestic point of the absorbance spectra of the *trans* and *cis* isomer (refer to Supplementary Information Figs. 43–45).

Concurrently, the wavelength-dependent photogeneration of the ketene moiety **K2** was investigated by irradiating samples of **K1** in DCM with an identical number of photons at each wavelength and determining the conversion to **K2** via $^1$H-NMR spectroscopy. These results are presented overlayed with the absorbance spectrum in Fig. 2B. It was found that the photogeneration of **K2** from $\alpha$-diazo ketone had a reaction quantum yield of $\phi_{375} = 0.17 \pm 0.02$ at 375 nm – a wavelength at which the azobenzene isomerisation was minimal. At wavelengths $\lambda > 550$ nm, the $\alpha$-diazo ketone **K1** does not absorb light and no formation of **K2** was observed. Importantly, the redshifted tetrachloride azobenzene moiety is essential for $\lambda$-orthogonal activation, as the non-substituted azobenzene shows minimal photoisomerization at wavelength $\lambda > 550$ nm and significant photoswitching at 390 nm (refer to Supplementary Information Figs. 40–42).

**Product analysis and polymer endgroup modification.** Having confirmed the orthogonal activation wavelengths of the two chromophores, we next investigate the product formed by the

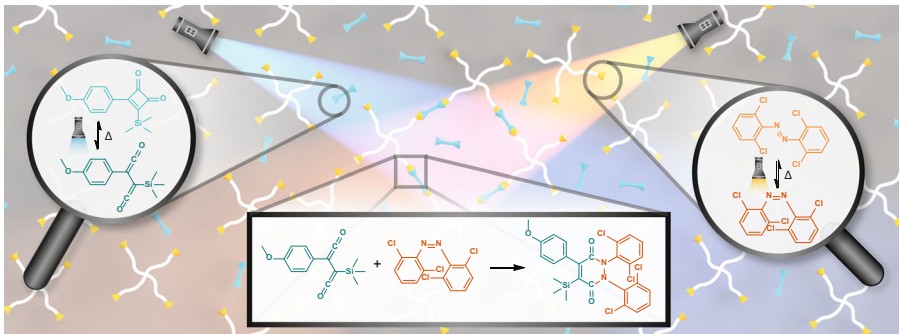

**Fig. 1 Schematic reaction overview.** Two-colour activation of a stable cyclobutenedione (UV light) and redshifted *ortho*-substituted tetrachloride azobenzene (orange light) enables formation of ligated product in polymer crosslinking.

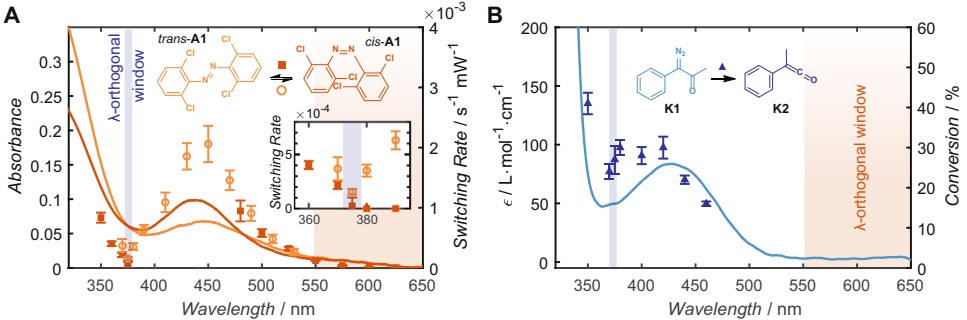

**Fig. 2 Orthogonal activation windows. A** Wavelength dependent rates of photoisomerization of **A1**, normalised to the incident light intensity and overlayed with the absorbance spectra of *trans*-**A1** (dark orange) and *cis*-**A1** (light orange). Switching rates were determined from exponential fits to 310 nm absorbance during irradiation. Error bars indicate variation in least squares fit. Inset highlights switching rates in the $\lambda$-orthogonal region from 355 to 395 nm. **B** Wavelength dependent reaction quantum yield of photoactivated ketene (**K2**) formation. Conversions determined by $^1$H-NMR spectroscopy of **K1** to **K2** after irradiation with $(3.8 \pm 0.3) \times 10^{18}$ photons of various wavelengths from a 20 Hz tuneable OPO. Error bars determined from standard deviation in pulse energy during irradiation.

reaction of *cis*-**A1** and the photogenerated ketene **K2**, in a one-pot reaction triggered by two-colour irradiation (Fig. 3). For these measurements, *trans*-**A1** and **K1** were dispersed in DCM and irradiated with a custom, two colour apparatus employing a tuneable laser and LED (refer to Supplementary Information Section 1.11). The results are presented in Fig. 3C. The photoproduct **P1** was isolated via reverse phase HPLC in good yields (79%) and fully characterised (refer to Supplementary Information Figs. 27–31). We find that the wavelengths determined from the action plot studies, although providing essential guidance, had to be adjusted. This is due to the change in photophysical properties of **A1** in the presence of **K1** associated with the shift in refractive index of the one-pot solution compared to pure DCM. If the UV wavelength was too short, unwanted *trans-cis* isomerisation of **A1** occurred, whereas longer wavelengths resulted in the switching rate of *cis-trans* isomerisation exceeding that of *trans-cis* with red light, and no *cis*-**A1** was produced. Consequently, $\lambda_1 = 380$ nm and $\lambda_2 = 650$ nm were found to be the ideal combination of wavelengths to independently address *cis*-**A1** and **K2**, respectively. Whilst a tuneable laser was vital in identifying these wavelength regions, subsequent studies revealed that the orthogonality was maintained when commercial LEDs ($\lambda_{1,max} = 385$ nm and $\lambda_{2,max} = 625$ or 650 nm) were employed (refer to Supplementary Information Figs. 53–55).

We further investigated the potential of our dual-colour photochemical system in polymer endgroup modification, towards polymer crosslinking for 3D printing. Specifically, we prepared a polyethylene glycol containing the photo reactive azobenzene endgroup (**A3**) by esterification of a monomethyl polyethylene glycol ($M_n = 2000$ g/mol) with the diazenyl benzoic acid (refer to Supplementary Information Fig. 21). The successful synthesis of **A3** was confirmed by the shift in $^1$H-NMR due to the presence of the tetrachloride azobenzene endgroup and the substitution pattern matching with the simulated mass spectrum of **A3** in size exclusion chromatography mass spectrometry (SEC-MS) (refer to Supplementary Information Fig. 53). To compare the NMR resonances of the photoligation between **A3** with **K1**, we also prepared the respective azobenzene small molecule **A2**, which contains a methyl ester, and synthesised the photo product **P2** (refer to Supplementary Information Figs. 32–34). The endgroup modification of **A3** with **K1** was monitored via SEC-MS and NMR spectrometry, and the results are presented in Fig. 3D. After irradiation with $\lambda_{1,max} = 385$ nm and $\lambda_{2,max} = 625$ nm, the ligation product **P3** (blue) was formed, indicated by the shift of the molecular distribution to lower retention volumes compared to the **A3** (orange). When only a single wavelength is used (grey), no shift, and therefore no increase in molecular mass, is observed, underpinning the orthogonality of our photochemical system. Similar results are obtained in the NMR measurements, where the chemical resonances of the isolated small molecule photoproduct **P2** are compared with the polymer endgroup modification **P3** in the case where both, or each individual, wavelengths are utilised (refer to Supplementary Information Fig. 54). The matching chemical resonances of **P2** only appear after irradiation with both wavelengths at 385 and 625 nm, and no product was observed when only one wavelength is employed. Ultimately, we tested the polymer coupling between PEG-azobenzene **A3** and a triethylene glycol ketene **K5** to determine the effect of the ligation. The SEC-analysis (refer to Supplementary Information Fig. 60) clearly shows an increase in molecular weight without the appearance of

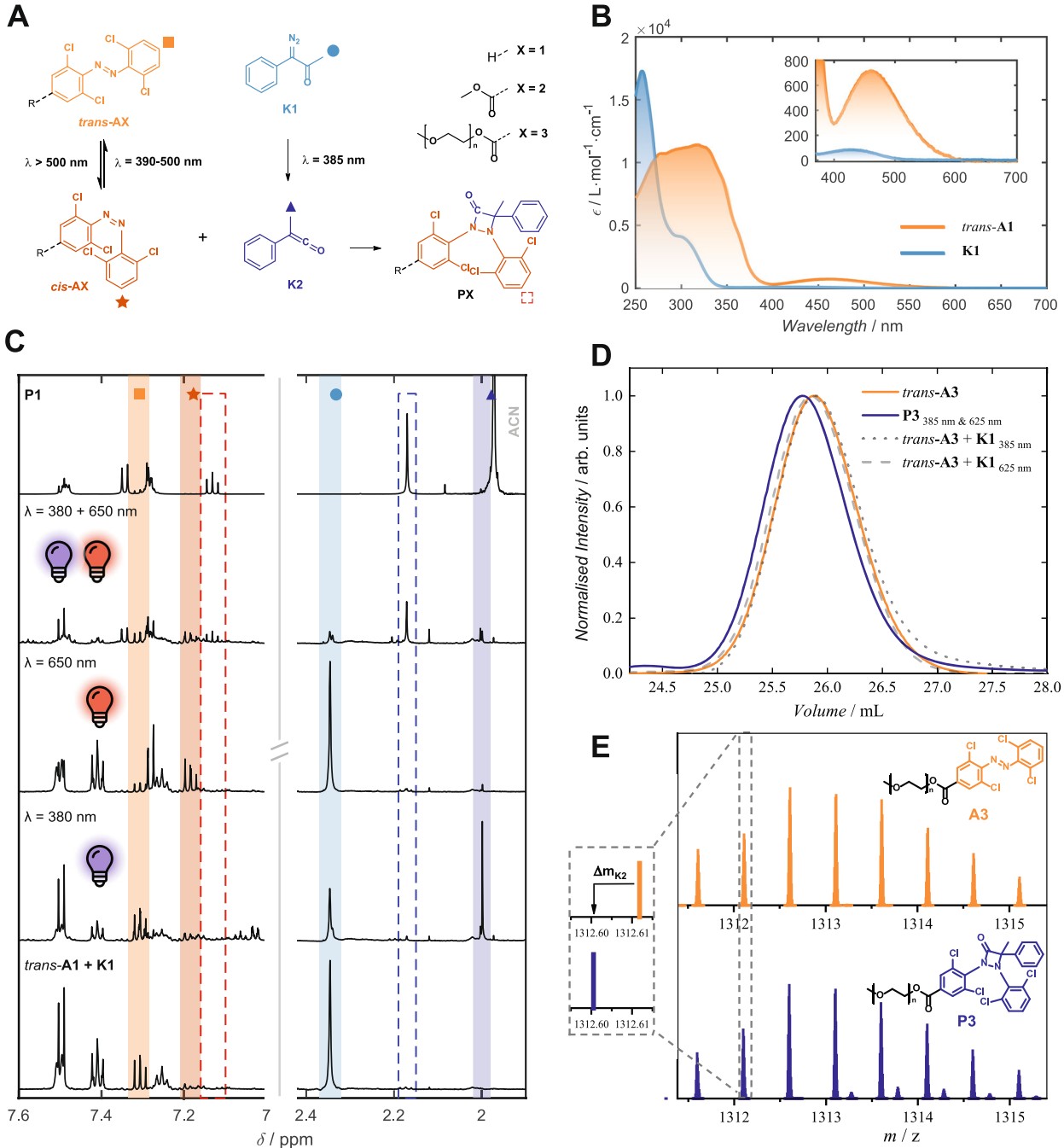

**Fig. 3 Two-colour photoligation system. A** Reaction scheme incorporating the *ortho*-substituted tetrachloride azobenzene (**A1**), which undergoes photoisomerisation from *trans* to *cis* under orange/red light activation. Under blue light irradiation, the photoactivated ketone (**K1**) transforms into the ketene (**K2**) which will undergo a thermal reaction with *cis*-**A1** to produce the ligation product **P1**. **B** Molar extinction coefficients of *trans*-**A1** and **K1** showing the potential orthogonal window ~400 nm. **C** ¹H-NMR spectra of unirradiated reaction mixture of *trans*-**A1** and **K1** (bottom), after irradiation with 650 nm only, 380 nm only and 650 nm and 380 nm, for comparison with isolated product **P1** (top). **D** Size-exclusion chromatography of **A3** and after irradiation with **K1** using 385 nm and 625 nm forming **P3**. Control experiments were conducted using only one wavelength, respectively. **E** Comparison of SEC-MS spectra of **A3** with the spectra after the irradiation of **A3** and **K1** with 385 and 625 nm forming **P3**. The isotopic pattern allows direct comparison of the weight gain upon endgroup modification.

any shoulder towards the starting distribution indicating quantitative ligation.

Upon end-group modification, the polymer distribution of **A3** is expected to show a mass change corresponding to the molecular mass of **K2**. Since the molecular weight of **K2** ($M = 132.0575$ g/mol) equals three times the molecular weight of ethylene glycol repeating unit ($M = 132.0786$ g/mol) the mass change is minor. With size exclusion chromatography coupled with high resolution mass spectrometry (SEC-HRMS) it is still possible to determine the change of isobaric masses by comparing their isotopologues (Fig. 3E, zoom-in). A representative section of the obtained mass distribution is displayed (refer to Supplementary Information Figs. 55–57) with the mass increase of $\Delta m_{K2}$ schematically indicated. Additionally, the pattern of **A3** is fully

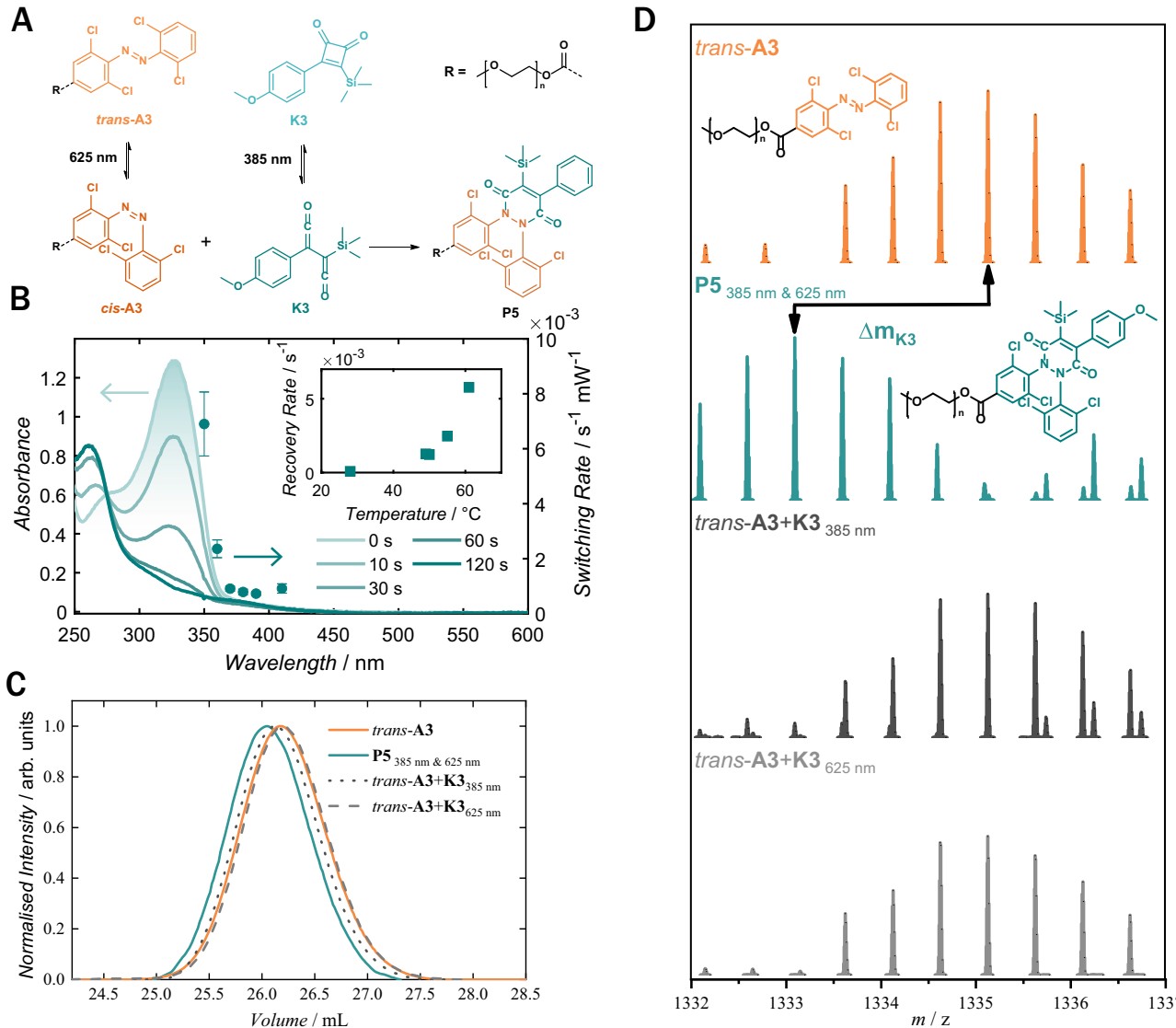

**Fig. 4 Photoreversible ketene coupling. A** Reaction mechanism of the two-colour light activation between **A3** and **K3**; only upon exposure to 385 nm and 625 nm the bond formation takes place resulting in **P4** or **P5** respectively. **B** In situ absorbance spectrum of **K3** in chloroform during exposure to 350 nm, 16 mW laser irradiation showing fast switching to the ketene **K3** (left axis) overlayed with the wavelength dependent switching rate (right axis). Inset highlights the temperature-dependent thermal recovery rate of **K3**. Switching rates were determined from exponential fits to 310 nm absorbance during irradiation. Error bars indicate variation in least squares fit. **C, D** SEC and SEC-MS data of **A3** (10 mmol/L in chloroform) before irradiation (orange) and after 15 min irradiation with 385 nm and 625 nm yielding **P5** (turquoise). Control experiments using only one wavelength, 385 nm or 625 nm respectively, are depicted in grey.

converted into **P3** with no unreactive starting material detectible after two colour irradiation, indicating the high selectivity of this ligation reaction.

**Photoreversible ketene for dual-colour coupling**. While we have achieved the dual-colour activation for the coupling reaction, the photo-generation of activated ketene **K2** is non-reversible, which may lead to the formation of side products due to the reactive nature of the ketene (refer to Supplementary Information Figs. 46, 47)[30,31]. To address this potential drawback, we designed a bisketene moiety **K3**, photogenerated from a stable cyclobutenedione, which can thermally revert back to the cyclobutadione in the dark (Fig. 4A)[32]. A series of experiments were conducted to investigate the reactivity of the cyclobutenedione **K3** with the tetrachloride azobenzene **A2**, under the same two-colour irradiation conditions explored above. First, the small molecule reaction between *trans*-**A2** and **K3** was examined in a

batch reaction setup. The product **P4** was purified via reverse phase HPLC, and its structure was fully confirmed by $^1$H-NMR, $^{13}$C-NMR, and 2D NMR spectroscopy, together with LCMS (refer to Supplementary Information Figs. 35–39). The two-wavelength initiated coupling was further applied to polymer endgroup modification, and the reaction of **A3** and **K3** was confirmed by SEC and SEC-MS (Fig. 4C, D). Inspection of the SEC traces (Fig. 4C) reveals a decrease in the retention volume of the polymer product, corresponding to an increase in molecular mass (orange to turquoise), confirming the formation of the polymer product **P5**. In control experiments where only one wavelength was used, we observed no product formation (dotted grey lines). The successful dual-wavelength endgroup modification is further confirmed by comparing the SEC-MS spectra of **A3** and **P5** (Fig. 4D), which shows the mass increase of **K3** after irradiation with 385 nm and 625 nm. Additionally, the isotopic pattern of the SEC-HRMS of **A3** and **P5** compared to their respective theoretically expected *m/z* values shows a mass deviation of 0.38-0.67

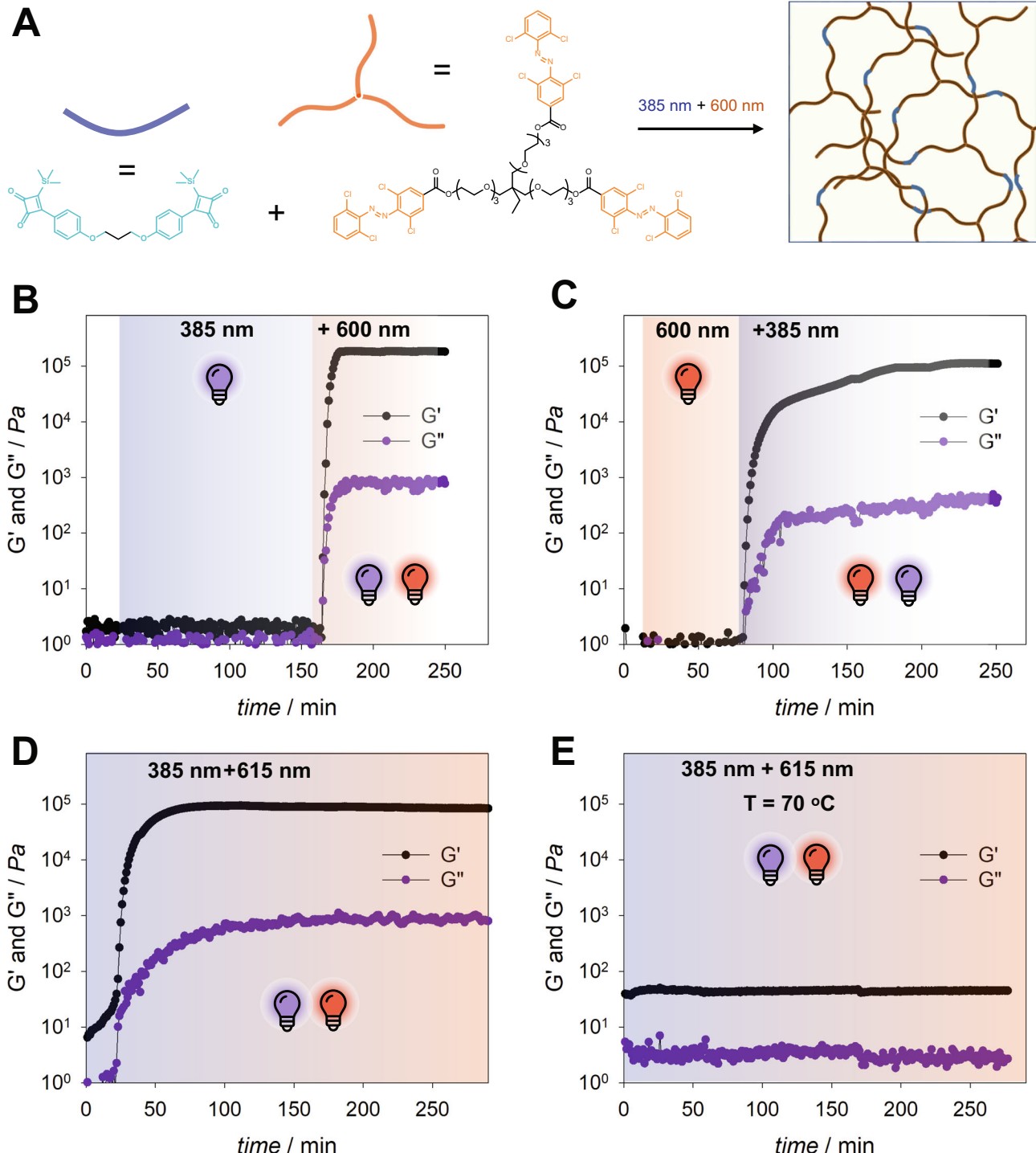

**Fig. 5 Dual-colour initiated polymer crosslinking. A** Representation of the crosslinking process between a biscyclobutenedione and a three-arm PEG functionalised with *ortho*-substituted tetrachloride azobenzene under two-colour irradiation. Rheological investigation of the crosslinking process under different irradiation conditions, showing the evolution of G′ and G″ values as the function of time: **B** the resin was irradiated first with UV light (385 nm), followed by red light (600 nm), **C** the resin was irradiated with red light, followed by UV light, at certain time interval UV light was switched off, halting the increase in G′ value, **D** the resin was irradiated by both red and UV lights simultaneously, and **E** the resin was heated to 70 °C and irradiated with both colours.

ppm, confirming the exact expected molecular composition (refer to Supplementary Information Fig. 59).

**Dual-colour initiated polymer crosslinking.** To assess the potential of our photochemical system for dual-colour 3D

printing, we formulated a resin containing a linear biscyclobutenedione **K4** and a three-arm PEG containing *ortho*-substituted tetrachloride azobenzene endgroup **A4** (Fig. 5A). The light-induced crosslinking was investigated by rheology specifically by following the evolution of the storage (G′) and loss (G″) moduli of the mixture under specific irradiation conditions[33,34]. As seen

from Fig. 5B, when the resin was irradiated with $\lambda_{max} = 385$ nm light, no change in either G' or G'' values was observed. When red light (600 nm) irradiation was added, an increase in both G' and G'' values was observed, indicating the formation of a solid network (Fig. 5B). Similarly, red light (600 nm) alone did not initiate crosslinking, but a combination of both wavelengths (385 nm and 600 nm) initiated crosslinking (Fig. 5C and D). It is noted that the sequence of irradiation from UV light to red light resulted in a faster crosslinking kinetics compared to the irradiation sequence starting with red light. We further demonstrate the temporal control over the two-colour crosslinking process by switching off one irradiation wavelength (Fig. 5C), and immediately halting the crosslinking. When the resin was irradiated with both wavelengths at elevated temperature (70 °C), crosslinking did not occur (Fig. 5E), suggesting that significant thermal reversion of the bisketene **K4** and *cis*-azobenzene **A4** occurred at elevated temperature (refer Supplementary Information Figs. 46 and 52). We further demonstrated the crosslinking on a material scale, using two LEDs to initiate the crosslinking of a resin, composed of a solution of biscyclobutenedione ($c = 0.3$ M) and three-arm PEG linker ($c = 0.3$ M) in DMF. In our experiments, solid gels formed only when a combination of two light colours are shone on the resin (refer to the Supplementary Movies 1–3).

## Discussion

Dual-colour 3D printing techniques require two photochemical transformations working in tandem to produce a single ligated product, and each of the process is initiated by discrete wavelengths. To date, reports on such photochemical systems are scarce. Commonly among the contemporary dual-colour photochemical systems, such as those reported by Regehly et al.[21] and our team[24] is the use of one non-reversible photoreaction. Our system uniquely requires the combination of two fully reversible photoactivations, where the product is formed exclusively by the irradiation with two wavelengths. In addition, the switching of the *ortho*-substituted tetrachloride azobenzene can be triggered by red light ($\lambda \geq 550$ nm), providing high penetration depth for the curing process. A detailed investigation of the photoreactivity of each process facilitated identification of the complementary activation wavelengths for the photoswitching of cyclobutenedione and azobenzene moieties. These findings were subsequently translated into polymer endgroup modification and materials crosslinking. Importantly, the two-colour ligation is highly selective and efficient. These findings have potential in advanced dual-colour printing techniques for fabrication of soft materials with localised variation in chemical and physical properties.

## Methods

**Synthetic procedures**. Detailed synthetic procedures are described in the Supplementary Information and are accompanied by reaction schemes and NMR characterisations figures.

**In-situ Absorbance**. For in-situ absorbance measurements during switching an Ocean Optics DH-MINI Deuterium-Tungsten-Halogen lamp was coupled via optic fibres (P400-025-SR) to an Ocean Optics FLAME-T-UV-VIS spectrometer, sensitive from 200 to 850 nm, via a cuvette holder. The cuvette holder was situated on a temperature-controlled stage to facilitate temperature-dependent switching experiments. An Opolette 355 tuneable OPO, emitting 5 ns pulses from 210-2400 nm at a repetition rate of 20 Hz, was directed onto a side of a quartz fluorescence cuvette perpendicular to the UV-Vis apparatus. Azobenzene **A1** samples were prepared in DCM with a concentration of 143 $\mu$mol L$^{-1}$ at 25 °C unless noted otherwise and photoreversible ketene (**K3**) samples were prepared in chloroform with a concentration of 52 $\mu$mol L$^{-1}$. Spectra were recorded every 10 s (50 ms integration time, 5 scan average) and processed in Matlab®.

**Ketene action plot measurements**. Samples of photoactivated ketene (**K1**) were prepared in CD$_2$Cl$_2$ with a concentration of 3.1 mmol L$^{-1}$. In all, 0.6 mL of solution was placed in an optically flat glass crimp vial for irradiation from below. The

light source was an Opotek Opolette 355 OPO, producing 7 ns, 20 Hz pulses with a flattop spatial profile. The output beam was expanded to 6 mm diameter to ensure it is large enough to uniformly irradiate the entire sample volume. The beam then passes through an electronic shutter and is directed upwards using a UV silica right angle prism. The laser energy deposited into the sample was measured above the aluminium block before and after experiments using a Coherent EnergyMax thermopile sensor (J-25MB-LE) to adjust the energy to deliver a constant photon count of $(3.8 \pm 0.3) \times 10^{18}$ photons at each wavelength.

**NMR two-colour studies**. Samples of *trans*-**A1** and **K1** (1.3 mmol L$^{-1}$, 1:2.3 equiv.) were prepared in CD$_2$Cl$_2$ and 0.6 mL was placed in an NMR tube for irradiation. UV light from Opotek Opolette 355 OPO was delivered from below as detailed above. The visible light irradiation was achieved with a 10 W LED ($\lambda_{max} = 650$ nm) mounted 3 cm from the side of the NMR tube. The irradiation time was controlled by an external shutter regulated with an Arduino-board. $^1$H-NMR spectra were recorded after 60 minutes of irradiation. For the measurement with 385 nm and 650 nm, the sample was left in the dark for 8 h after irradiation to achieve maximum product formation.

**Batch reactions**. The synthesis of **P1**, **P2** and **P4** was conducted in a batch setup using a 1.5 mL crimped vial. The ketene (**K1**, **K3**, 2.0-3.0 eq) and the azobenzene (*trans*-**A1**, *trans*-**A2**, 1.0 eq) were dissolved in dichloromethane/chloroform (15-25 mmol L$^{-1}$) and irradiated for 10-30 min simultaneously with 385 nm (1 A, 20 V, 2 cm distance) from one side 625 nm (2.1 A, 22 V, 2 cm distance) from the other side. The LEDs were cooled using a stream of air and a fan. After completion the solvent was removed under reduced pressure and submitted to reverse phase HPLC (Acetonitrile:Water gradient from 5:95 to 100:0). The product was obtained as a yellow/orange oil (67–84% yield).

**Rheology studies**. Rheological experiments were measured using an Anton Paar Physica rheometer (MCR 302e) with a plate-plate configuration. The lower plate is made of quartz and the upper plate is made of stainless steel with a diameter of 25 mm. The LED light sources were placed below the quartz plate (refer Supplementary Information Fig. 6). In a typical experiment, a solution (50 $\mu$L) of 3-arm PEG-azobenzene (0.2 M) and ketene **K4** (0.3 M) in acetonitrile was placed on the lower plate and the upper plate was lowered to a measurement gap of 0.2 mm. The measurement was started by applying a 1% strain with a frequency of 0.1 Hz on the sample.

## Data availability

The authors declare that the data supporting the findings of this study are available within the article and its Supplementary Information Files. Extra data are available from the corresponding author upon request.

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

## Acknowledgements

C.B.-K. acknowledges funding by the Australian Research Council (ARC), in the context of a Laureate Fellowship underpinning his photochemical research programme. C.B.-K. acknowledges continued funding by the Deutsche Forschungsgemeinschaft (DFG, German Research Council) under Germany's Excellence Strategy for the Excellence Cluster '3D Matter Made to Order' (Exc-2082/390761711). C.B.-K. and J. P. B. acknowledge support via an ARC Discovery project targeted at red-shifting photoligation chemistry as well as continued key support from the Queensland University of Technology (QUT). The size exclusion chromatography hyphenated with mass spectrometry data and rheology data reported herein were obtained at the Central Analytical Research Facility (CARF).

## Author contributions

S.L.W. undertook the wavelength-dependent photoactivation experiments and small molecule studies. L.L.R contributed to the synthesis and conducted small molecule and polymer endgroup characterisations. J.A. contributed to the synthesis. V.X.T. contributed to the synthesis of small molecules, preparation of the polymers, and conducted rheology studies. V.X.T., J.P.B and C.B.-K. conceptualised the study. J.P.B. and C.B.-K. were responsible for writing the grants on which the study is based (ARC Laureate C.B.-K., ARC Discovery C.B.-K. and J.P.B.). All authors contributed to manuscript preparation and editing.

## Competing interests

The authors declare no competing interests.
