## [Peer review file · Nature Communications]

REVIEWER COMMENTS

Reviewer #1 (Remarks to the Author):

This manuscript reports on the synthesis, characterization, and application of photo-active two-component systems (azo/ketene cycloaddition, a well-established reaction originating in the work of Saudinger). Each component is individually activated at certain wavelengths.

Using state-of-the-art equipment, the authors have evaluated the optimal irradiation conditions. Thorough control and reference experiments have been documented.

Throughout the manuscript there is a real mess between K1/K2 and the corresponding ketenes and photo-generated ketenes (even in the supporting information). The authors must thoroughly distinguish between the precursors and the ketenes.

Generally the authors claim that achieving a photostationary state of the Z-azo component and the (reversible) generation of a ketene causes the desired cycloaddition, which in terms of a polymerisation, leads to crosslinking. Here, I am not confident that this is a particular benefit for a 3D polymerisation process. However, this procedure may be rather useful to produce selected polymers with specific topologies. I would appreciate if the authors could be more specific in terms of applications of their approach.

Reviewer #2 (Remarks to the Author):

The authors present a photo-induced cycloaddition reaction which only proceeds under irradiation with 2 wavelengths at once. This is done using a substituted azobenzene, which can be activated by red light but shows little isomerization at ~380 nm as one component. The other reaction partner is a ketene which is formed under UV light irradiation (2 different precursors, one with an irreversible ketene formation, and one where the reaction is thermoreversible). Both react in a cycloaddition reaction. The wavelength dependency of the reaction is mapped and its success and selectivity under various irradiation conditions is probed. This is also done using PEG-conjugates of the azo-benzene. Finally, resin formation which only occurs when both wavelengths are coupled in, is demonstrated using a divalent ketene and a trivalent azobenzene derivative.

I find this study quite fascinating and think this is a significant step forward in this field. A dual wavelength induced reaction with both upstream reactions being reversible is extremely promising. The study is conducted well, and I support the conclusions drawn by the authors. What I think could use some improvement is the presentation. There seem to be some mistakes in some graphs (see comments below) but even without them it is sometimes too hard to understand what is presented in detail. Especially for a broad audience as expected for Nature Comm the authors should reassess if it is possible to present the information in a more straight forward and intuitive way.

From the side of experiments, I would be highly interested in seeing a polymer-polymer coupling reaction. This is always a great benchmark for such a coupling reactions. Even if this shows no quantitative coupling, this isn't a severe disadvantage for this study in my opinion. It would just show the limits of what can be achieved with this particular system.

Also in order to use this reaction for 3D printing, I don't think it's fast enough at this point. Again, I don't see this as a problem for the current study but a short coming that should be addressed in the manuscript (or disproven by an actual 3d printing experiment).

Overall, I think this manuscript could be accepted after major revision.

Comments:

Scheme 1 looks very nice but could be more informative. For instance, for someone who isn't familiar with this kind of cycloaddition, it's relatively hard to make out what is reacting with each other in what way. Perhaps adding a simplified reaction mechanism (similar to the one in the SI) that ignores all the residues that are not essential to the cycloaddition would help. After all, there would be enough space for that. I think especially when targeting a broad audience this would be helpful. Scheme 1 will be the item which every reader will look at first so it should be easier to understand the chemistry based on it.

Figure 1A: I am not sure what the insert is. Is this switching rate? If so the axis title is missing and it's unintuitive that the axis numbering is on the left side (abs. in the full plot) and not factored (10^{-3}) as the right y axis. Also, the 550 nm, which marks the beginning of the orange area in the full plot should also be in the insert (maybe including the orange area?)

Other than that it's hard to make out what exactly happens at 375nm. A zoom in on that region would be much more interesting. At the moment it looks like this negligible switching at 275 is in the same order of magnitude as what happens at 550 nm.

"As expected, this wavelength corresponds to the isosbestic point of the absorbance spectra of the trans and cis isomer." Is this referring to Figure S39/40? If so please refer to it in the text.

“Whilst a tuneable laser was vital in identifying these wavelength regions, subsequent studies revealed that the orthogonality was maintained when commercial LEDs ($\lambda_{1,max} = 385$ nm and $\lambda_{2,max} = 625$ or 650 nm) were employed.” please link to data (in SI?)

Something is wrong in Figure 2. If the peak assignment in the NMR is correct then the trans A1 is doing nothing under red light and isomerization happens at 380 nm. On the “blue” side: K1a is unreactive under 380 nm and reacts at 650 nm. The scheme and the discussion states it reverse. Please clarify. Is the order just messed up?

Naming of substitution in Figure 2A is confusing. 1 should be “X=1”, 2 should be “X=2” and then PEG-X2 should be “X=PEG” after that logic and sample name for conjugate would then be cis-APEG ?

Anyway, it took far too long to figure out what the authors mean. There is certainly a better, more intuitive way to do this.

Figure 2D: the shift in the SEC looks nice. It would be really interesting to see what happens when both molecules are conjugated to a polymer. Then the coupling could be followed by SEC and would give a nice insight into how good this reaction is performing. This would be even more interesting for the final system shown in Figure 3A! A relatively easy experiment would be taking the cross linker from the resin synthesis with 2 equivalents of the PEG conjugated azobenzene and look at the product in SEC. This would show how quantitative the reaction works when confronted with such a sterically difficult problem.

Figure 3E and following: please provide an axis with numbers for the free floating mass spec. Also the x axis doesn't seem to have the right numbers. The spectra should be an isotopic pattern, but the assigned m/z values have only a 0.01 change? Same problem in the respective SI graph.

Then, it seems that the reaction is not quantitative from the NMR (Neither the isomerization nor the ketene formation and consequently also not the cycloaddition. The full MS (Figure S51) I think should be provided in higher resolution (regarding the plot, thinner lines etc) to see the remaining precursor distribution if there is one. All distributions should be assigned as well

Figure 3D: what is the second isotopic pattern when PEGtrans A2 is irradiated at 385 nm?

Some minor spelling mistakes: “formaulated” on page 8; a “.” after the reference in the introduction. Some double space's.

As stated, overall I support this study and think it will be a very nice piece for Nature Comm after some revisions.

Reviewer #1

Comments: *This manuscript reports on the synthesis, characterization, and application of photo-active two-component systems (azo/ketene cycloaddition, a well-established reaction originating in the work of Saudinger). Each component is individually activated at certain wavelengths.*

Using state-of-the-art equipment, the authors have evaluated the optimal irradiation conditions. Thorough control and reference experiments have been documented.

Response: We thank the reviewer for their positive overview of our equipment and experimental methods. We do not claim to be the first to report the azobenzene/ketene reaction. The novelty in this work is in redshifting the activation wavelengths and generating a photoreversible ketene, thus opening the door for reversible, two colour activated, bond formation. In recognition of the prior work undertaken in developing the underlying reaction, we have included the reference suggested by the reviewer.

Comments: *Throughout the manuscript there is a real mess between K1/K2 and the corresponding ketones and photo-generated ketenes (even in the supporting information). The authors must thoroughly distinguish between the precursors and the ketenes.*

Response: We have taken on board the comments of the reviewer, in combination with the comments of Reviewer 2, who both expressed confusion around the labelling of structures. To simplify matters we have labelled each structure with a unique identifier and removed the 'a' and 'b' suffixes from the precursor and photogenerated ketene. We hope this has resolved the technical issue.

Comments: *Generally the authors claim that achieving a photostationary state of the Z-azo component and the (reversible) generation of a ketene causes the desired cycloaddition, which in terms of a polymerisation, leads to crosslinking. Here, I am not confident that this is a particular benefit for a 3D polymerisation process. However, this procedure may be rather useful to produce selected polymers with specific topologies. I would appreciate if the authors could be more specific in terms of applications of their approach.*

Response: We thank the reviewer for raising their concerns regarding the application of our work. However, would like to offer a perspective on its viability for 3D printing. As has already been shown by Hecht and coworkers, two colour printing systems can rapidly increase printing speeds while offering mild curing conditions (compared to tightly focussed femtosecond laser beams), please refer to this publication: <https://www.nature.com/articles/s41586-020-3029-7>. Such approaches require that the curing of the photoresist is triggered by two orthogonal colours light. To the best of our knowledge, only one resist exists that fulfils this requirement (refer to the noted *Nature* paper above), yet there is no resist based on two orthogonally addressable photoswitches, who when activated react to form a two-colour gated covalent bonds. Whether our specific resist can be employed in two colour gated printing devices, remains to be explored. We have edited the introduction to clarify the point noted by the reviewer.

Reviewer #2

Comments: *The authors present a photo-induced cycloaddition reaction which only proceeds under irradiation with 2 wavelengths at once. This is done using a substituted azobenzene, which can be activated by red light but shows little isomerization at ~380 nm as one component. The other reaction partner is a ketene which is formed under UV light irradiation (2 different precursors, one with an irreversible ketene formation, and one where the reaction is thermoreversible). Both react in a cycloaddition reaction. The wavelength dependency of the reaction is mapped and its success and selectivity under various irradiation conditions is probed. This is also done using PEG-conjugates of the azo-benzene. Finally, resin formation which only occurs when both wavelengths are coupled in, is demonstrated using a divalent ketene and a trivalent azobenzene derivative.*

I find this study quite fascinating and think this is a significant step forward in this field. A dual wavelength induced reaction with both upstream reactions being reversible is extremely promising. The study is conducted well, and I support the conclusions drawn by the authors. What I think could use some improvement is the presentation. There seem to be some mistakes in some graphs (see comments below) but even without them it is sometimes too hard to understand what is presented in detail. Especially for a broad audience as expected for Nature Comm the authors should reassess if it is possible to present the information in a more straight forward and intuitive way.

Response: We thank the reviewer for their overall positive assessment of the study. We agree that reversible activation of both components constitute a promising and significant step forward in the field. We are extremely grateful for the detailed nature in which the reviewer has assessed our work and we have taken their comments onboard, to which we respond in detail below. We hope the reviewer will find the presentation of the revised manuscript improved and more broadly accessible to the readership of *Nature Communications*.

Comments: *From the side of experiments, I would be highly interested in seeing a polymer-polymer coupling reaction. This is always a great benchmark for such a coupling reactions. Even if this shows no quantitative coupling, this isn't a severe disadvantage for this study in my opinion. It would just show the limits of what can be achieved with this particular system.*

Response: We thank the reviewer for their suggestion, and we have undertaken additional experiments to further validate the efficiency of the two-colour induced coupling. Therefore, we synthesised an oligomer, derived from triethylene glycol, containing the photo-active diazoketone moiety (**K5** in the SI) and conducted the two-colour ligation experiment as well as the control experiments with the PEG-azobenzene **A3**. We have included the results in the SI and added an additional explanation to the main text: "Ultimately, we tested the polymer coupling between PEG-azobenzene **A3** and a triethylene glycol ketene **K5** to determine the effect of the ligation. The SEC-analysis (refer to supporting information, Figure S60) clearly shows an increase in molecular weight without the appearance of any shoulder towards the starting distribution indicating quantitative ligation."

Comments: *Also in order to use this reaction for 3D printing, I don't think its fast enough at this point. Again, I don't see this as a problem for the current study but a short coming that should be addressed in the manuscript (or disproven by an actual 3d printing experiment).*

Response: The reviewer makes an excellent point, i.e. the rate at which the two independently addressable photoswitches revert after excitation. It is entirely possible that the current system reverts too slowly during printing and further improvements are required. To adapt the system for 3D printing, which includes adaptations with regard to the viscosity of the resist mixture among others, access to, e.g., a Xolography printer is required. However, currently only a few printers exist in the world capable of two colour light sheet printing and we unfortunately do not have access to one, yet we seek to explore collaborative opportunities in this space. In response to the reviewer's comment, we have made note of the slow reversion in the revised manuscript.

Comments: *Overall, I think this manuscript could be accepted after major revision.*

Response: We thank the reviewer for their confidence in the manuscript and trust they will find out revisions satisfactory.

Comments: *Scheme 1 looks very nice but could be more informative. For instance, for someone who isn't familiar with this kind of cycloaddition, its relatively hard to make out what is reacting with each other in what way. Perhaps adding simplified reaction mechanism (similar to the one in the SI) that ignores all the residues that are not essential to the cycloaddition would help. After all, there would be enough space for that. I think especially when targeting a broad audience this would be helpful. Scheme 1 will be the item which every reader will look at first so it should be easier to understand the chemistry based on it.*

Response: We thank the reviewer for their comments and have now included a reaction scheme of the chemical reaction taking place within the confines of Scheme 1.

Comments: *Figure 1A: I am not sure what the insert is. Is this switching rate? If so the axis title is missing and its un-intuitive that the axis numbering is on the left side (abs. in the full plot) and not factored (10⁻³) as the right y axis. Also, the 550 nm, which marks the beginning of the orange area in the full plot should also be in the insert (maybe including the orange area?)*

Other than that its hard to make out what exactly happens at 375nm . A zoom in on that region would be much more interesting. At the moment it looks like this negligible switching at 275 is in the same order of magnitude as what happens at 550 nm.

Response: As per the reviewer's suggestion, we have modified the inset of Figure 1 (which previously showed the switching rates on a logarithmic scale) to show a zoomed in region around 375 nm, where the switching is minimal. The reviewer is correct in their assessment that the switching rate at 375 nm is comparable to the switching rate at 550nm. The difference between these two wavelength regions, leading to the desired orthogonality, is that at 375 nm the equilibrium strongly favours the *trans*-azobenzene and very little of the reactive *cis*-azobenzene is produced. We have included a graph of the equilibriums in the SI (Figure S49) to highlight this very important point.

Comments: *“As expected, this wavelength corresponds to the isosbestic point of the absorbance spectra of the trans and cis isomer.” Is this referring to Figure S39/40? If so please refer to it in the text.*

Response: The isobestic point is visible in Figure 1, however a much clearer example can be seen in Figure S39/40. As per the reviewer’s suggestion, a reference to the relevant figures in the SI has been added the manuscript.

Comments: *“Whilst a tuneable laser was vital in identifying these wavelength regions, subsequent studies revealed that the orthogonality was maintained when commercial LEDs (I1,max = 385 nm and I2,max = 625 or 650 nm) were employed.” please link to data (in SI?)*

Response: We have added a reference to Figures S49 and S50 where the orthogonality is demonstrated with two LED’s and confirmed by size exclusion chromatography and NMR spectroscopy.

Comments: *Something is wrong in Figure 2. If the peak assignment in the NMR is correct than the trans A1 is doing nothing under red light and isomerization happens at 380 nm. On the “blue” side: K1a is unreactive under 380 nm and reacts at 650 nm. The scheme and the discussion states it reverse. Please clarify. Is the order just messed up?*

Response: We are extremely grateful to the reviewer for bringing the matter to our attention. Indeed, the order was incorrect and the mislabelled spectra have now been corrected.

Naming of substitution in Figure 2A is confusing. 1 should be “X =1”, 2 should be “X=2” and then PEG-X2 should be “X=PEG” after that logic and sample name for conjugate would then be cis-APEG ? Anyway, it took far too long to figure out what the authors mean. There is certainly a better, more intuitive way to do this.

Response: As confusion around labelling was raised by both reviewers, we have carefully attended to the matter and simplified the structure labelling. Each structure has now been given a unique identifier, eliminating the reference to ‘PEG’ structures. We trust that the modified naming convention is more intuitive to follow.

Comments: *Figure 2D: the shift in the SEC looks nice. It would be really interesting to see what happens when both molecules are conjugated to a polymer. Then the coupling could be followed by SEC and would give a nice insight into how good this reaction is performing. This would be even more interesting for the final system shown in Figure 3A! A relatively easy experiment would be taking the cross linker from the resin synthesis with 2 equivalents of the PEG conjugated azobenzene and look at the product in SEC. This would show how quantitative the reaction works when confronted with such a sterically difficult problem.*

Response: We thank the reviewer for their helpful suggestion and have undertaken the suggested experiment. Specifically, we have synthesized a PEG containing the diazoketene endgroup (termed **K5** in the

SI), and conduct the two-colour induced ligation with the PEG containing the azobenzene endgroup (**A3**). Our result indicates a very efficient coupling, with the SEC trace of the product clearly shifting towards larger molecular weights. We have included an additional brief discussion in the manuscript to describe the additional result: “Ultimately, we tested the polymer coupling between PEG-azobenzene **A3** and a triethylene glycol ketene **K5** to determine the effect of the ligation. The SEC-analysis (refer to supporting information, Figure S60) clearly shows an increase in molecular weight without the appearance of any shoulder towards the starting distribution indicating quantitative ligation.”

Comments: *Figure 3E and following: please provide an axis with numbers for the free floating mass spec. Also the x axis doesn't seem to have the right numbers. The spectra should be an isotopic pattern, but the assigned m/z values have only a 0.01 change? Same problem in the respective SI graph.*

Response: We thank the reviewer for the very valuable comment and agree that the presented spectra 2E might have been confusing without further description. Therefore, we included the adjusted isotopic pattern and added the requested axis. In addition, we now included the explanation for the small m/z change during the end-group modification:

“Upon end-group modification, the polymer distribution of **A3** is expected to show a mass change corresponding to the molecular mass of **K2**. Since the molecular weight of **K2** ($M = 132.0575$ g/mol) equals three times the molecular weight of ethylene glycol repeating unit ($M = 132.0786$ g/mol) the mass change is minor. With size exclusion chromatography coupled with high resolution mass spectrometry (SEC-HRMS) it is still possible to determine the change of isobaric masses by comparing their isotopologues (Figure 2E, zoom-in). A representative section of the obtained mass distribution is displayed (refer to supporting information, Figure S55-57) with the mass increase of Δm_{K2} schematically indicated.”

Comments: *Then, it seems that the reaction is not quantitative from the NMR (Neither the isomerization nor the ketene formation and consequently also not the cycloaddition).*

Response: This is a good observation and we agree that neither the switching nor the ligation reaction is quantitative. In the NMR experiments we didn't intent to drive the reaction to full conversion to prevent side reactions. Additionally, the isomerization, similar to other isomerization reactions, has its equilibrium at 70-80% *cis*-isomer. Furthermore, in Section 2.8 of the supporting information we determined the isolated yield of the photoreaction (79%) to show that the reaction is not quantitative. Despite this, by implementing the reaction on the materials scale, we demonstrate that a fully quantitative reaction is not required for our intended application.

Comments: *The full MS 8Figure S51 I think should be provided in higher resolution (regarding the plot, thinner lines etc) to see the remaining precursor distribution if there is one. All distributions should be assigned as well*

Response: Figure S51 was adjusted and replaced as requested. For the assignments of the residual distributions, we want to point the reviewer's attention to the following Figures S52, 53. Here, we provided detailed information regarding all observed distribution based on simulated and matched spectra. Generally, we want to point out that mass spectrometry cannot be used to obtain quantitative information. Depending

on the ionisation strength of starting material and product small impurities can appear severe if they are charged easily but do not represent the real amount.

Comments: *Figure 3D: what is the second isotopic pattern when PEGtrans A2 is irradiated at 385 nm?*

Response: The isotopic pattern emerging when the polymer is irradiated with 385 and 625 nm and with only 385 nm is a side reaction of the reactive ketene with the polymer. Since it does not match with the calculated pattern of the product or the starting material, the structure is unknown. However, we do not believe the side reaction is problematic or surprising since we do observe side reactions on the small molecule scale already. Furthermore, we cannot estimate the amount, which could be very minor, of the impurities since mass spectrometry is not a quantitative method, as described previously.

Comments: *Some minor spelling mistakes: "formaulated" on page 8; a "." after the reference in the introduction. Some double space's.*

Response: Thank you for bringing this to our attention. The error has been corrected.

Comments: *As stated, overall I support this study and think it will be a very nice piece for Nature Comm after some revisions.*

Response: Again, we thank the reviewer for their positive assessment and trust they will find out revisions satisfactory.

REVIEWERS' COMMENTS

Reviewer #1 (Remarks to the Author):

The authors have considered the suggestions of both reviewers and have accomplished a more readable version of the manuscript.

From my side, the manuscript is now ready for publication.

Reviewer #2 (Remarks to the Author):

The authors have addressed all comments in a satisfying way.

The revised manuscript can be published in Nature Communications.